# Molecular mechanism of allosteric activation of the CRISPR ribonuclease Csm6 by cyclic tetra-adenylate

Liyang Du 🆔, Qinwei Zhu & Zhonghui Lin 🆔 ✉

## Abstract

**Type III CRISPR systems are innate immune systems found in bacteria and archaea, which produce cyclic oligoadenylate (cOA) second messengers in response to viral infections. In these systems, Csm6 proteins serve as ancillary nucleases that degrade single-stranded RNA (ssRNA) upon activation by cOA. In addition, Csm6 proteins also possess cOA-degrading activity as an intrinsic off-switch to avoid degradation of host RNA and DNA that would eventually lead to cell dormancy or cell death. Here, we present the crystal structures of Thermus thermophilus (Tt) Csm6 alone, and in complex with cyclic tetra-adenylate ($cA_4$) in both pre- and post-cleavage states. These structures establish the molecular basis of the long-range allosteric activation of TtCsm6 ribonuclease by $cA_4$. $cA_4$ binding induces significant conformational changes, including closure of the CARF domain, dimerization of the HTH domain, and reorganization of the R-$X_{4-6}$-H motif within the HEPN domain. The cleavage of $cA_4$ by the CARF domain restores each domain to a conformation similar to its apo state. Furthermore, we have identified hyperactive TtCsm6 variants that exhibit sustained $cA_4$-activated RNase activity, showing great promise for their applications in genome editing and diagnostics.**

**Keywords** Type III CRISPR; Cyclic Oligoadenylate; Ancillary Nuclease; Allosteric Activation; Csm6
**Subject Categories** Microbiology, Virology & Host Pathogen Interaction; Structural Biology

## Introduction

The type III CRISPR-Cas system features a unique cyclic oligoadenylate (cOA) signaling mechanism that confers prokaryotes an additional layer of protection against viral RNA (Athukoralage and White, 2021, 2022; Kazlauskiene et al, 2017; Koonin and Makarova, 2018; Molina et al, 2020; Niewoehner et al, 2017; Rouillon et al, 2018). Upon foreign RNA invasion, the Cas10 protein (also known as Csm1 or Cmr2) in the CRISPR-Cas III complexes synthesizes cOA molecules, which are comprised of three to six 3´-5´ linked AMPs (Athukoralage and White, 2021, 2022; Han et al, 2018; Jia et al, 2019; Kazlauskiene et al,

2017; Koonin and Makarova, 2018; Molina et al, 2020; Niewoehner et al, 2017; Rouillon et al, 2018; Sofos et al, 2020; You et al, 2019). These molecules act as second messengers that can activate a variety of downstream promiscuous ancillary nucleases to degrade RNA and/or DNA (Kazlauskiene et al, 2017; McMahon et al, 2020; Niewoehner et al, 2017; Rostol et al, 2021; Samolygo et al, 2020; Zhu et al, 2021). Among these, the best-characterized ones are the Csm6 (Cas subtype Mtube 6) and Csx1 (cardiac-specific homeobox 1). Structural studies have revealed that the functions of Csm6/Csx1 proteins are primarily mediated by two functionally distinct domains: the CARF (CRISPR-associated Rossman-fold) domain that specifically binds the cOA second messenger, and the HEPN (higher eukaryotes and prokaryotes nucleotide) domain which catalyzes RNA degradation (Garcia-Doval et al, 2020; Makarova et al, 2020; Molina et al, 2019; Niewoehner and Jinek, 2016).

While the cOA-dependent indiscriminate degradation of RNA and/or DNA results in the clearance of invaders, the prolonged activation of this signaling can lead to cell dormancy or cell death (Rostol and Marraffini, 2019). Recent studies have identified a group of cOA-degrading enzymes called ring nucleases that dedicate to bisect the cOA molecules into two linear products, providing an off-switch regulation for the cOA signaling (Athukoralage et al, 2019; Athukoralage et al, 2020a; Athukoralage et al, 2020b; Athukoralage et al, 2018; Brown et al, 2020; Du et al, 2023; Molina et al, 2022; Molina et al, 2021). The majority of ring nucleases are CARF domain-containing proteins, which include the Crn1 (CRISPR ring nuclease 1) proteins Sso2081 and Sso1393 from *S. solfataricus* (Athukoralage et al, 2018), as well as Sis0811 (Molina et al, 2021) and Sis0455 (Molina et al, 2022) from *S. islandicus*. However, there are also CARF-unrelated ring nucleases that are encoded by some archaeal viruses and bacteriophages, such as Crn2/AcrIII-1 (anti-CRISPR III-1) (Athukoralage et al, 2020a) and Crn3/Csx3 (Athukoralage et al, 2020b; Brown et al, 2020). In addition, in the absence of standalone ring nucleases, certain Csm6 proteins have evolved to intrinsically degrade their cOA activators similarly to the standalone ring nucleases (Athukoralage et al, 2019; Garcia-Doval et al, 2020; Jia et al, 2019; Smalakyte et al, 2020). In this case, the CARF domains of these Csm6 proteins function as both a cOA sensor and a ring nuclease.

So far, self-limiting cOA-dependent Csm6 proteins have been described from many species, such as *T. thermophilus* (Athukoralage et al, 2019), *T. onnurineus* (Jia et al, 2019), *E. italicus* (Garcia-Doval et al, 2020), and *S. thermophilus* (Smalakyte et al, 2020). These Csm6 proteins display diversity in amino acid sequence (<15% identity) and utilize distinct types of cOA as second

College of Chemistry, Fuzhou University, 350108 Fuzhou, China. ✉E-mail: zhonghui.lin@fzu.edu.cn

messengers. For instance, ToCsm6 (Jia et al, 2019) and TtCsm6 (Athukoralage et al, 2019) utilize cyclic tetra-adenylate ($cA_4$), whereas EiCsm6 (Garcia-Doval et al, 2020) and StCsm6 (Smalakyte et al, 2020) employ cyclic hexa-adenylate ($cA_6$). To date, various structures of Csm6 proteins have been determined, including TtCsm6 alone (Niewoehner and Jinek, 2016), EiCsm6 complexed with non-degradable $cA_6$ mimic (Garcia-Doval et al, 2020), and ToCsm6 either alone or bound to $cA_4$ in different catalytic states (Jia et al, 2019). These structures have provided insight into the mechanism by which the CARF domain recognizes and cleaves cOA activators, as well as how the HEPN domain catalyzes ssRNA degradation, yet the mechanism of cOA-mediated allosteric activation of the HEPN ribonuclease still remains an open question.

In this study, we present the crystal structures of TtCsm6 alone, and in complex with $cA_4$ in both pre- and post-cleavage states. These structures show that $cA_4$-binding induces significant global and local conformational changes throughout the CARF, HTH, and HEPN domains of TtCsm6. The $R-X_{4-6}-H$ motif within the HEPN domain adopts a catalytically incompetent conformation prior to $cA_4$ binding and after $cA_4$ cleavage, whereas $cA_4$ binding induces a reorganization of the motif, thereby activating the ribonuclease. Together with extensive biochemical analyses, our study thus establishes the structural basis of cOA-mediated allosteric activation of the type III CRISPR ancillary ribonuclease Csm6.

## Results

### Crystal structure of TtCsm6/$cA_4$ complex in post-cleavage state

The recombinant TtCsm6 protein was purified from bacterial cells. In the gel mobility-based RNA cleavage assay, the purified TtCsm6 protein could efficiently cleave ssRNA in time- and $cA_4$-dependent manners (Fig. 1A). In addition, in the LC/MS-based $cA_4$ cleavage assays, TtCsm6 could catalyze the conversion of $cA_4$ into 5′-OH-ApA-2′, 3′-cyclic phosphate ($A_2 > P$) (Fig. 1B; Appendix Fig. S1). These results thus confirm that TtCsm6 is a $cA_4$-dependent self-limiting ribonuclease (Niewoehner et al, 2017).

To investigate the mechanism of $cA_4$ recognition by TtCsm6, we next co-crystallized the protein with $cA_4$, and determined the structure at a resolution of 1.8 Å by molecular replacement, using the structure of apo-TtCsm6 (PDB: 5FSH) as a template (Table 1). The overall structure is similar to that of apo-TtCsm6 (Niewoehner and Jinek, 2016), with a root-mean-square deviation (RMSD) value of 1.0 Å (Fig. 1C). The two TtCsm6 monomers dimerize in an X-shape, and each monomer contains an N-terminal CARF domain, an intermediate helix-turn-helix (HTH) domain and a C-terminal HEPN domain (Fig. 1C). The refined electron density map clearly illustrates that $cA_4$ in the CARF domain was cleaved into two $A_2 > P$ molecules (Fig. 1D), suggesting that the structure was determined in the post-cleavage state.

### Mechanism of $cA_4$ recognition

Compared to the apo form, the CARF domain bound by $A_2 > P$ undergoes a significant movement in the mobile loop spanning residues from 165 to 179, which encloses the two $A_2 > P$ molecules into the catalytic pocket of CARF domain (Fig. 1E). In the absence

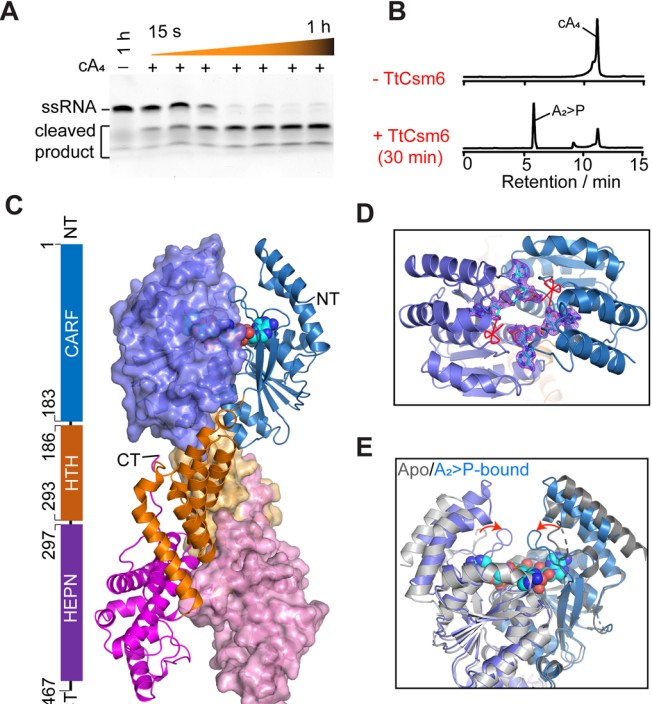

**Figure 1. Crystal structure of TtCsm6 in complex with $A_2 > P$.**

(A) Denatured polyacrylamide gel analysis of TtCsm6 ribonuclease activity against 6-carboxyfluorescein (FAM)-labeled ssRNA. (B) HPLC spectra of $cA_4$ cleavage by TtCsm6. (C) Overall structure of TtCsm6/$A_2 > P$ complex. Left: Schematic representation of domain composition of TtCsm6. Right: Cartoon representation of TtCsm6/$A_2 > P$ complex, with one monomer overlaid with 50% transparent surface. The CARF, HTH, and HEPN domains are colored in blue, orange and magenta, respectively. The N and C termini are indicated as NT and CT. The $A_2 > P$ molecules are shown in cyan spheres. (D) Top view of the TtCsm6 CARF domain bound by $A_2 > P$, superimposed with the 2Fo-Fc electron density map contoured at 1.0 σ. Cleavage sites are indicated by red scissors. All structural figures in this study, unless otherwise specified, follow the same color and labeling schemes. (E) Structural alignment of CARF domains bound to $A_2 > P$ (blue) with its apo form (gray). Conformational changes upon ligand binding are highlighted by red arrows. Source data are available online for this figure.

of ligand, this loop is partially disordered and adopt an open conformation, suggesting an induced-fit mechanism for ligand binding. Similar mechanisms have been observed previously in the standalone ring nucleases, such as Sso2081 (Du et al, 2023), Sis0455 (Molina et al, 2022), and AcrIII-1 (Athukoralage et al, 2020a).

The two $A_2 > P$ molecules bind to a composite site across the TtCsm6 dimer interface (Fig. 1D). Sequence alignments reveal that the CARF domain of TtCsm6 has minimal sequence conservation with that of the characterized Csm6 proteins such as ToCsm6 and EiCsm6. Instead, it shares high sequence similarity with Crn1 standalone ring nucleases such as Sso2081 and Sis0455 (Appendix Fig. S2). These include the residues in the two motifs adjacent to β-strand 1 (G58, T59, and S60) and β-strand 4 (T133, G135, and K137) of TtCsm6, corresponding to the previously described signature motif-I and motif-II of ring nucleases (Fig. 2A; Appendix Fig. S2) (Du et al, 2023; Makarova et al, 2020). These results indicate that the mechanism by which TtCsm6 binds and

**Table 1. Data collection and refinement statistics.**

| | TtCsm6$^{K137A}$ | TtCsm6$^{Y161A}$/cA$_4$ > P | TtCsm6/A$_2$ > P |
|---|---|---|---|
| **Data collection** | | | |
| Space group | P 21 21 2 | P 43 21 2 | P 1 21 1 |
| Cell dimensions | | | |
| *a, b, c* (Å) | 94.89, 205.57, 59.10 | 116.24, 116.24, 156.71 | 58.24, 75.12, 103.69 |
| α, β, γ (°) | 90, 90, 90 | 90, 90, 90 | 90, 95.9, 90 |
| Resolution (Å) | 29.55–2.41 (2.50–2.41)* | 37.61–2.89 (2.99–2.89) | 29.68–1.81 (1.88–1.81) |
| *R*$_{merge}$ (%) | | | |
| *I* / σ*I* | 15 (1.8) | 18.0 (2.2) | 9.9 (1.7) |
| Completeness (%) | 99.87 (99.93) | 99.88 (100.00) | 99.54 (99.43) |
| Redundancy | 12.9 (13.5) | 25.4 (26.3) | 6.1 (5.0) |
| **Refinement** | | | |
| Resolution (Å) | 29.55–2.41 | 38.85–2.89 | 29.68–1.81 |
| No. of reflections | 45,567 | 24,689 | 80,743 |
| *R*$_{work}$/*R*$_{free}$ | 0.229/0.236 | 0.222/0.290 | 0.195/0.201 |
| No. of atoms | 7149 | 7280 | 7809 |
| Macromolecules | 7112 | 7246 | 7269 |
| Ligand/ion | 2 | 1 | 2 |
| Water | 35 | 34 | 540 |
| *B*-factors | 75.41 | 73.32 | 37.42 |
| Macromolecules | 75.47 | 73.56 | 36.98 |
| Ligand/ion | 116.21 | 56.55 | 38.21 |
| Water | 60 | 65.08 | 43.06 |
| R.m.s. deviations | | | |
| Bond lengths (Å) | 0.009 | 0.010 | 0.018 |
| Bond angles (°) | 1.12 | 1.23 | 1.06 |
| Ramachandran plot (%) | | | |
| Favored/allowed/disallowed | 96.96/3.94/0.00 | 94.09/5.47/0.44 | 98.47/1.31/0.22 |

*Values in parentheses are for the highest-resolution shell.

cleaves cA$_4$ is largely conserved with the Crn1 standalone ring nucleases.

The adenine group of A1/A1′ accepts two hydrogen bonds from residues T81 and E83, while their 2′, 3′-cyclic phosphate is stabilized by residues T59, K137 and R172 (Fig. 2A,B). The T59A or S60A single mutation did not have a significant effect, but the T59A/S60A double mutation and K137A single mutation abrogated cA$_4$ cleavage (Fig. 2D), consistent with their conserved roles in cA$_4$ cleavage. Furthermore, alanine substitution of T81 but not E83 greatly suppressed the ring nuclease activity of TtCsm6 (Fig. 2D), indicating that T81 plays a dominant role in stabilizing the adenine group of A1/A1′. On the other hand, the adenine group of A2/A2′ is specifically recognized by R173 and E179 from the mobile loop, while the 2′-OH group is stabilized by N164 (Fig. 2A,C). Of note, the phosphate connecting A1 and A2 (or A1′ and A2′) closely contacts the phenolic hydroxyl group of Y161, which makes ~90° rotation in response to cA$_4$ binding (Fig. 2C). Despite not directly contacting the scissile phosphate or 2′-OH nucleophile, alanine substitution of either Y161 or N164 markedly diminished cA$_4$ cleavage (Fig. 2D). In addition, although the point mutation of

Y167A, R172A, R173A or E179A in the mobile loop did not trigger significant defects, the R172A/R173A double mutation caused ~50% reduction of cA$_4$ cleavage, and the Y167A/R172A/R173A triple mutation completely abolished cA$_4$ cleavage (Fig. 2D), underscoring the importance of the mobile loop in cA$_4$ binding.

## Identification of TtCsm6 variants with sustained cA$_4$-activated RNase activity

When cA$_4$ was pre-incubated with TtCsm6, we observed no stimulation of ssRNA degradation (Appendix Fig. S3), indicating that the ribonuclease activity of TtCsm6 could not be activated by the cleavage product of cA$_4$. This also suggests that the HEPN domain of A$_2$ > P-bound TtCsm6 is in a catalytically incompetent conformation. Therefore, we next sought to determine the structure of TtCsm6/cA$_4$ complex in the pre-cleavage state. To this end, we first conducted a microscale thermophoresis (MST) assay to test the cA$_4$ binding capability of the TtCsm6 mutants that were deficient in cA$_4$ cleavage, as described above. HPLC analysis revealed that cA$_4$ remained uncleaved after pre-incubation with both wild-type and

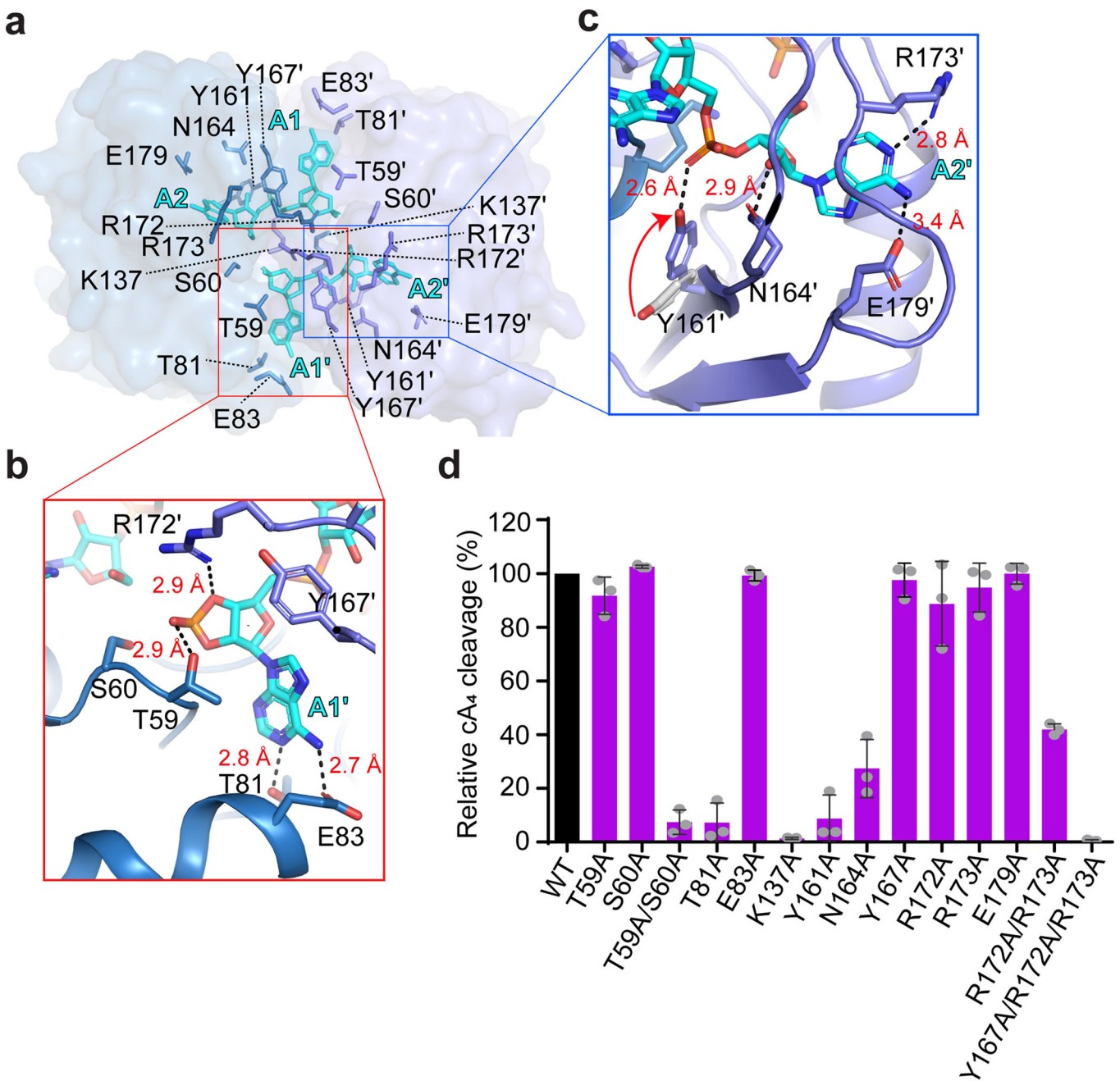

**Figure 2. The A$_2$ > P binding site within the TtCsm6 CARF domain.**

(A) The interaction between A$_2$ > P and TtCsm6 CARF domain, overlaid with 80% transparent surface. The key interacting residues are shown in the sticks and labeled. (B, C) Close-up views of A1/A1′ (B) and A2/A2′ (C) binding sites. Hydrogen bonds are indicated by dash lines, and the bond distance are labeled. Red arrow indicates the conformational change upon ligand binding. (D) Quantitation of LC-MS analyses for the cA$_4$ cleavage activity of wild-type (WT) TtCsm6 and its mutants. Values are means ± SD, $n = 3$ for technical replicates. Source data are available online for this figure.

mutant TtCsm6 for 15 min at room temperature in the MST binding conditions (Appendix Fig. S4). TtCsm6 exhibited robust binding affinity to cA$_4$, with a $K_D$ of approximately 15 nM (Fig. 3A). In comparison, the T59A/S60A mutant demonstrated approximately fivefold weaker binding than the WT (Fig. 3B). In addition, T81A was about 100-fold weaker (Fig. 3C), and K137A completely lost its ability to bind cA$_4$ (Fig. 3D), suggesting that these residues

mainly contribute to cA$_4$ binding. Furthermore, the Y161A and N164A mutants retained a tight binding affinity to cA$_4$ similar to the WT protein (Fig. 3D–F). Based on the results from cA$_4$ binding and cleavage assays, we propose that residues T81, and K137 are primarily responsible for cA$_4$ binding, while residues Y161 and N164 play a major role in cA$_4$ cleavage. In addition, residues T59 and S60, which coordinate both the scissile phosphate and the

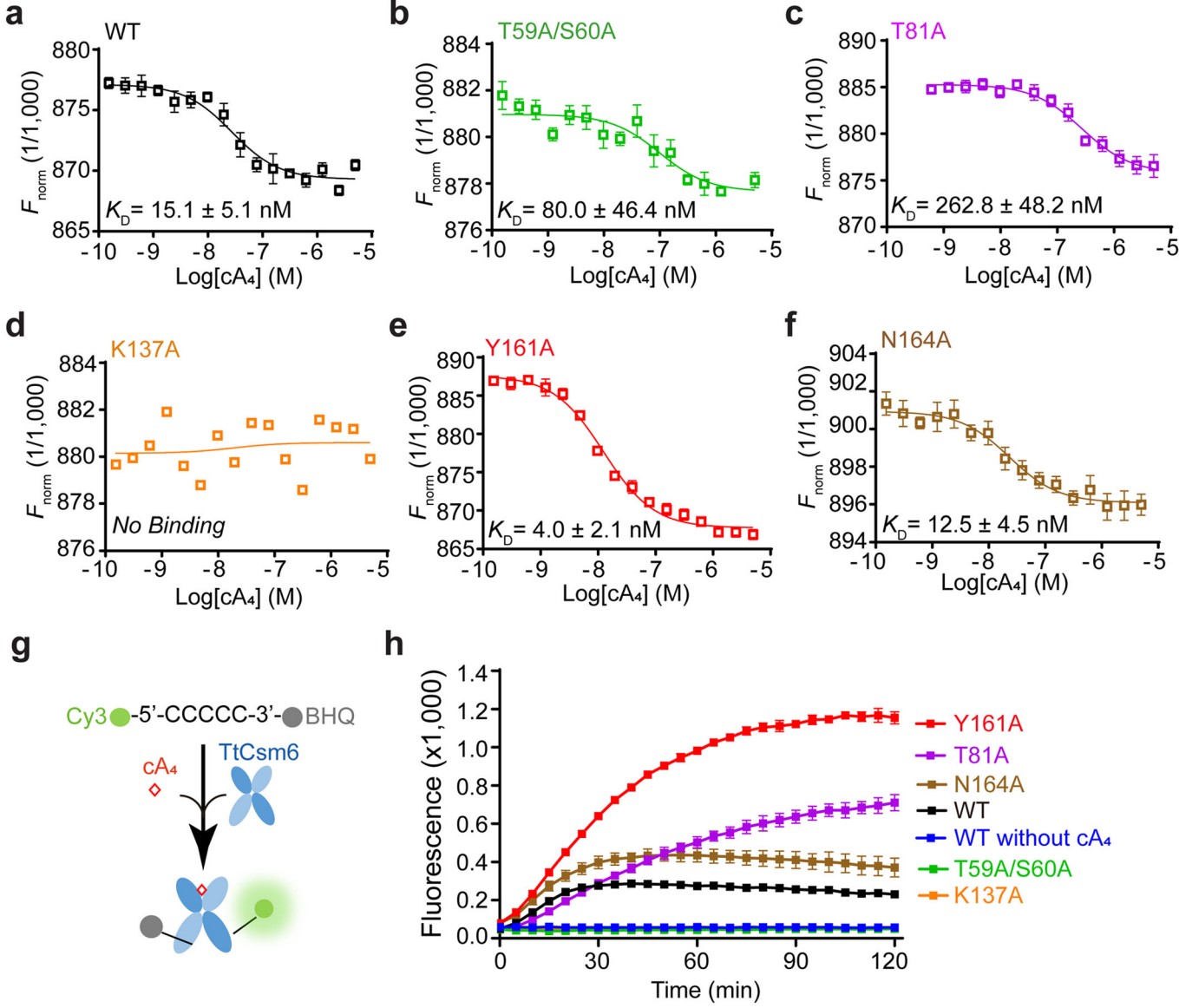

**Figure 3. Identification of TtCsm6 and variants with sustained cA$_4$-activated RNase activity.**

(A–F) The binding affinities of TtCsm6 and its mutants were evaluated by the microscale thermophoresis (MST) binding assay, using 50 nM of the labeled proteins and varying concentrations of cA$_4$. Values represent means ± SD, $n \geq 3$ for technical replicates. (G) Schematic illustration of the FRET-based assay used for determining TtCsm6 ribonuclease activity. (H) Kinetic analyses of ribonuclease activity for TtCsm6 and its mutants. Values are means ± SD, $n = 3$ for technical replicates. Source data are available online for this figure.

2´-OH nucleophile, may be required for both cA$_4$ binding and cleavage.

To further evaluate the ribonuclease activity of T81A, Y161A, and N164A, we next performed a FRET-based RNA cleavage assay (Fig. 3G). In agreement with the results of the gel-based assay, TtCsm6 alone did not induce a significant increase of the fluorescence intensity. However, in the presence of cA$_4$, we observed a gradual increase that reached a plateau within 30 min (Fig. 3H). Remarkably, the T81A, Y161A, and N164A mutations resulted in more significant and prolonged RNA degradation than the WT (Fig. 3H). In contrast, mutation of T59A/S60A or K137A completely eliminated RNA degradation activity (Fig. 3H).

It is intriguing that the T59A/S60A mutant exhibits only about a 5-fold weaker binding affinity than the wild-type (WT) yet is completely impaired in cA$_4$ cleavage and RNA degradation. In contrast, the T81A mutant displays approximately a 100-fold weaker cA$_4$ binding affinity than WT but surprisingly demonstrates more pronounced and sustained RNA degradation. We propose a possible explanation for this seemingly paradoxical result. Because T59 and S60 play a crucial role in positioning the 5´-phosphate and 2´-OH nucleophile, the T59A/S60A mutation likely disrupts the proper positioning of cA$_4$ within the CARF domain, potentially interfering with the allosteric activation of the HEPN RNase. This disruption, in turn, explains the lack of cA$_4$ cleavage and RNA degradation observed in this mutant. Conversely, the

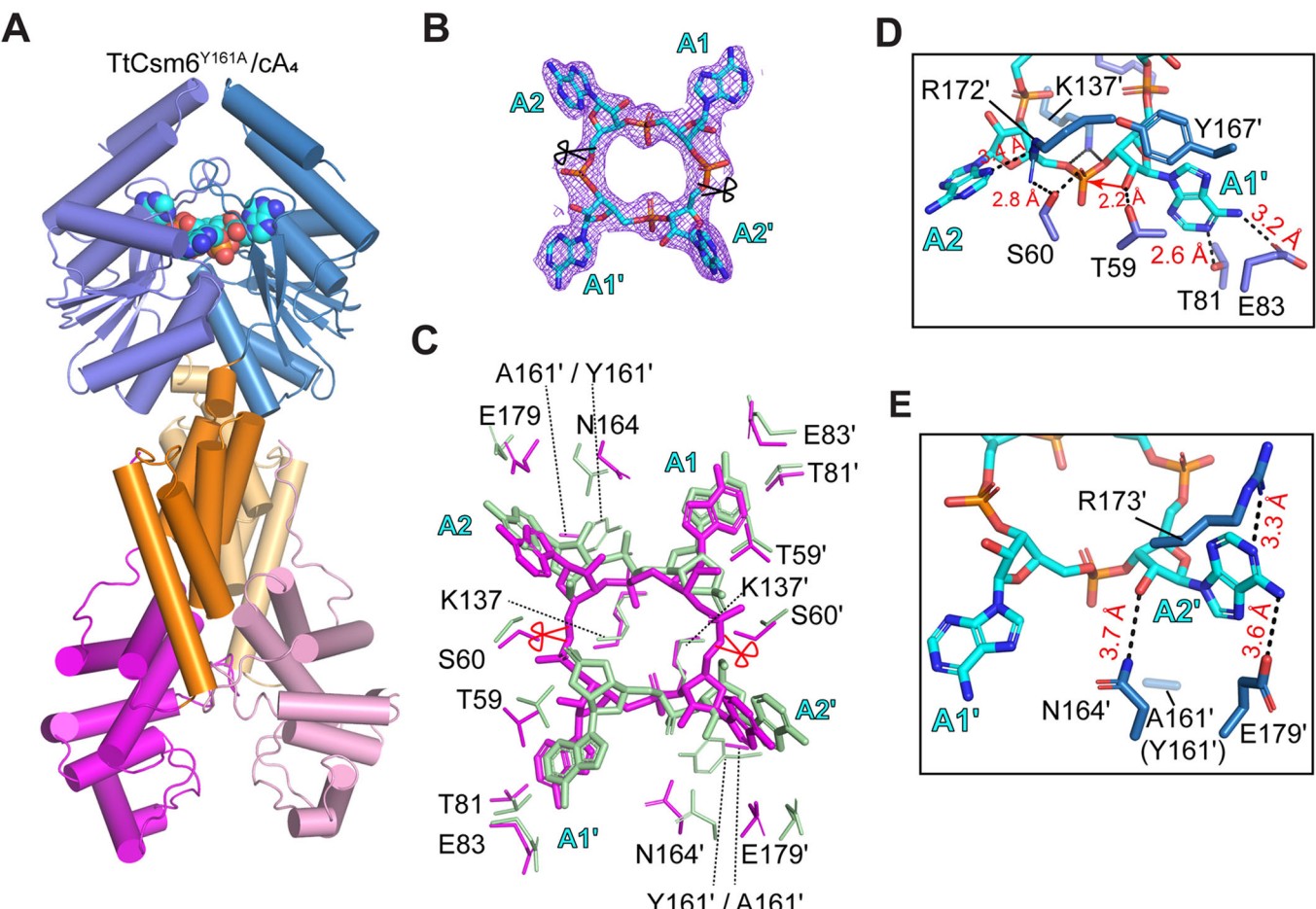

**Figure 4. Crystal structure of TtCsm6 in complex with cA₄.**

(A) Overall structure of the TtCsm6/cA₄ complex. α-helices are represented as cylinders, and cA₄ is in sphere. (B) Stick representation of cA₄ in the crystal structure, superimposed with the 2Fo-Fc electron density map contoured at 1.0 σ. (C) Comparison of cA₄ (magenta) and A₂ > P (green) binding sites within the TtCsm6 CARF domain. Key interacting residues are shown as sticks and labeled. The scissile sites within cA₄ are indicated with red scissors. (D, E) Close-up views of the interaction between cA₄ and TtCsm6 CARF domain. Hydrogen bonds are denoted by dash lines, and the bond distance is labeled. Red arrow indicates nucleophilic attack by the 2´-OH group.

T81A mutation, although it leads to a substantial reduction in cA₄ binding affinity, may not significantly alter the binding mode of cA₄ within the CARF domain. As a result, while this mutation initially causes a decrease in RNA degradation within the first 30 min of the reaction compared to the WT, over an extended timeframe, the T81A mutant demonstrates more significant and prolonged RNA degradation. The persistence of cA₄ binding within the CARF domain, albeit with reduced affinity, may account for this unexpected observation.

Taken together, these results identify TtCsm6 variants (particularly the Y161A mutant) with sustained cA₄-activated RNase activity, which show great promise for their applications in further structural studies as well as CRISPR-based genome editing and diagnostics.

## Structure of TtCsm6/cA₄ complex in pre-cleavage state

We next co-crystallized TtCsm6^Y161A with cA₄ and determined the structure at a resolution of 2.9 Å (Table 1). As expected, we observed a cA₄ molecule binding in the CARF domain (Fig. 4A), which remains

uncleaved as evidenced by its unambiguous electron density (Fig. 4B), indicating that the structure was determined in the pre-cleavage state. Superposition of the structure with that of apo and post-cleavage states yielded RMSD values of 3.5 Å and 2.7 Å, respectively (Appendix Fig. S5A), suggesting that significant conformational changes occur between the three different states, which will be discussed in detail in the following sections of this paper.

The overall binding mode of cA₄ exhibits some similarity to A₂ > P, but there are also noticeable differences (Fig. 4C). One significant variation is that, in this cA₄-bound structure, we were able to observe the coordination of both the scissile phosphate and the 2´-OH nucleophile, which are positioned by S60 and T59, respectively, resulting in an O2´-P-O5´ angle of 159° (Fig. 4C,D). This observation is consistent with the previous notion that cA₄ cleavage in CARF domain does not require deprotonation of attacking nucleophile or protonation of the leaving group via acid-base catalysis, but occurs due to steric factors that cause the ligand to adopt a conformation that enables in-line nucleophilic attacking (Athukoralage et al, 2018; Garcia-Doval et al, 2020; Jia et al, 2019).

In addition, the configuration of nonscissile phosphates also differs significantly from that of $A_2 > P$. Specifically, in $cA_4$, the $P = O$ bonds of the nonscissile phosphates point towards the bottom of the pocket, while in $A_2 > P$, these bonds make a 90° rotation and face away from the ring center (Fig. 4C). Furthermore, the contact of nonscissile phosphate with Y161 is abrogated due to the alanine substitution (Fig. 4E). After cleavage, the two $A_2 > P$ fragments separate with each other by ~3.4 Å compared to the positioning of $cA_4$. In response to these changes, nearly all of the $cA_4$ binding residues shift by 1.5-3.5 Å (Fig. 4C–E).

## Conformational changes of TtCsm6 during the $cA_4$ catalysis cycle

We observed both global and local structural changes of TtCsm6 during the $cA_4$ catalysis cycle (Fig. 5A; Movies EV1 and EV2). Upon $cA_4$ binding, the primary conformational change occurs in the CARF domain, where a mobile loop along with the two helices situated above the catalytic center switch from an open to a closed conformation, completely enclosing the $cA_4$ molecule within the catalytic center (Fig. 5A,B). It is worth noting that while the CARF domain bound by $A_2 > P$ also displays a closed conformation compared to the apo form, it is noticeably distinct from the conformation observed in the $cA_4$-bound state (Fig. 5B). It adopts an intermediate conformation between the apo and $cA_4$-bound states (Fig. 5A). Point mutation in the mobile loop, including Y167A, R172A, R173A, and E179A, abolished the ribonuclease activity of HEPN domain (Appendix Fig. S5B,C). These data strongly suggest that the mobile loop mediates the allosteric activation of HEPN ribonuclease by sensing $cA_4$ binding and cleavage in CARF domain.

Another prominent conformational change was observed in the HTH domain. In the $cA_4$-bound structure, the HTH domains of the two monomers move toward each other with a maximum shift of ~10 Å compared to the apo state (Fig. 5C; Movie EV1). This movement also causes the two HTH domains misalign along the dimer axis for approximately 5 Å (Fig. 5C). As a result, a new dimer interface is formed between helices 2 and 3 of the two HTH domains, involving residues V194, F198, E201, L202, E207, Q210, and Y214 (Fig. 5C). Together, these interactions generate a buried surface of about 360 Å². On the contrary, these interactions were not observed in the $A_2 > P$-bound structure, where the HTH domain adopts a conformation similar to that of the apo state (Fig. 5C).

We next investigated the structural changes in the HEPN domain. The HEPN domain of TtCsm6 contains a conserved $R-X_{4-6}-H$ motif (R415, N416, and H422). Consistent with the previous finding (Niewoehner and Jinek, 2016), point mutation of R415 A, N416A, or H422A abrogated the ribonuclease activity of TtCsm6 (Appendix Fig. S6A). In the previously reported structure of apo-TtCsm6, residues N416 and H422 are coordinated by a $Ni^{2+}$ ion (Appendix Fig. S6B) (Niewoehner and Jinek, 2016), which may cause a conformational distortion and hence generate an artifact. Coincidentally, in the structure of TtCsm6$^{K137A}$, which we initially determined to investigate the impact of K137A mutation on the structural integrity of TtCsm6 homodimer, no metal ions were detected in the HEPN domain (Table 1). The Ni-free structure of TtCsm6 is essentially identical to that of the Ni-bound one both in the overall structure and in the $R-X_{4-6}-H$ motif (Appendix

Fig. S6C,D). Thus, $Ni^{2+}$ ion-binding does not significantly change the active site arrangement of HEPN domain.

In comparison to the apo structure, $cA_4$ binding induces the HEPN domain of the two monomers move in opposite directions perpendicular to the dimer axis (Fig. 5A; Movie EV1). More importantly, we observed notable changes in the $R-X_{4-6}-H$ motif of HEPN domain. In the apo state, N416 and H422 of monomer-1 form hydrogen bonds with H422' and N416' of monomer-2, respectively, and the guanidine group of R415 is buried inside the domain, where it is stabilized by E332 (Fig. 5D). However, in the $cA_4$-bound structure, the hydrogen bonds between N416 and H422 are disrupted, causing them to separate with each other by a maximum distance of 9.4 Å (Fig. 5D). Moreover, R415 also disassociates from E332 and exposes its guanidine side chain to the surface of catalytic center (Fig. 5D). This remodeling positions the $R-X_{4-6}-H$ motif into a catalytically competent conformation. On the contrary, the conformation of HEPN domain including the $R-X_{4-6}-H$ motif in the $A_2 > P$-bound state is essentially identical to that of the apo state (Fig. 5D). This observation is consistent with the above results that the cleavage product of $cA_4$ does not activate the ribonuclease activity of TtCsm6.

## Discussion

The Csm6/Csx1 proteins are known as cOA-activated type III CRISPR-Cas ancillary nucleases. Some Csm6 enzymes, including TtCsm6, exhibit inherent HEPN-dependent ribonuclease activity even in the absence of cOA ligands (Kazlauskiene et al, 2017; Niewoehner et al, 2017; Niewoehner and Jinek, 2016). This indicates that their HEPN domains undergo a conformational equilibrium between activated and inactive states (Garcia-Doval et al, 2020), which poses a challenge in studying the mechanism of cOA-mediated allosteric activation of RNA degradation. In this study, we initially determined the crystal structure of TtCsm6 in complex with $cA_4$ in its post-cleavage state, uncovering the mechanism of how TtCsm6 recognizes and cleaves $cA_4$. Guided by this structure, we then identified a TtCsm6 variant Y161A, which displays a sustained $cA_4$-activated RNase activity without cleaving $cA_4$. This enabled us to subsequently determine the structure of TtCsm6 bound to $cA_4$ in a ribonuclease-competent conformation.

Based on the structures and biochemical data presented here, we propose a model to elucidate the mechanism of how $cA_4$ binding in the CARF domain of TtCsm6 allosterically activates its HEPN ribonuclease, as well as how $cA_4$ cleavage by the CARF domain abrogates this long-range (~70 Å) allosteric activation (Fig. 6; Movies EV1 and EV2). First, in the absence of $cA_4$, the key residues within the $R-X_{4-6}-H$ motif of HEPN domain are misaligned, H422 forms a hydrogen bond with N416, and R415 is buried inside the domain and stabilized by E332, rendering it in a ribonuclease-incompetent state (Fig. 5D). $cA_4$ binding to the CARF domain induces a transition of this domain from an open to a closed conformation (Fig. 6, step 1; Movie EV1). This conformational change subsequently promotes the dimerization of the following two HTH domains (Fig. 5C; Movie EV1), which in turn drives the movement of HEPN domain, freeing residues R415 and H422 to engage in 5′-phosphate stabilization and 2′-OH nucleophile activation (Fig. 5D; Movie EV1). While bound to $cA_4$,

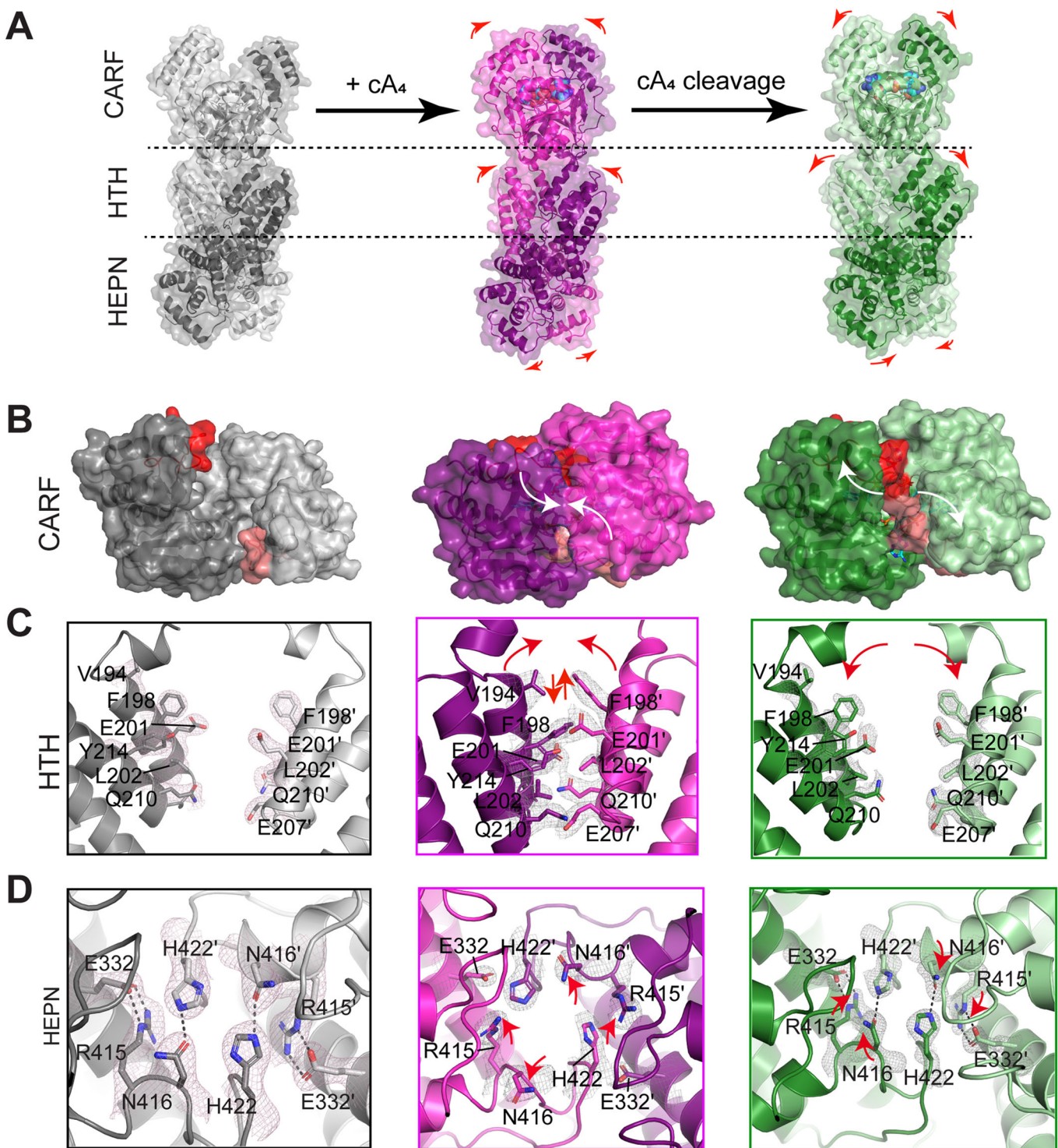

**Figure 5.  Structural comparison of TtCsm6 at different catalytic states.**

Apo state is shown in gray, cA₄-bound state in magenta and A₂ > P-bound state in green. (A) Global conformational changes upon cA₄ binding and cleavage. The structures are depicted with cartoons overlaid with 60% transparent surfaces. (B) Top view of the TtCsm6 CARF domains in different states, with the mobile loops (residue 165-179) highlighted in red. (C, D) Zoomed-in views of the TtCsm6 HTH (C) and HEPN (D) domains in different states. Key residues are shown in the sticks and labeled, superimposed with the 2Fo-Fc electron density map contoured at 1.0 σ. Major conformational changes are indicated by arrows.

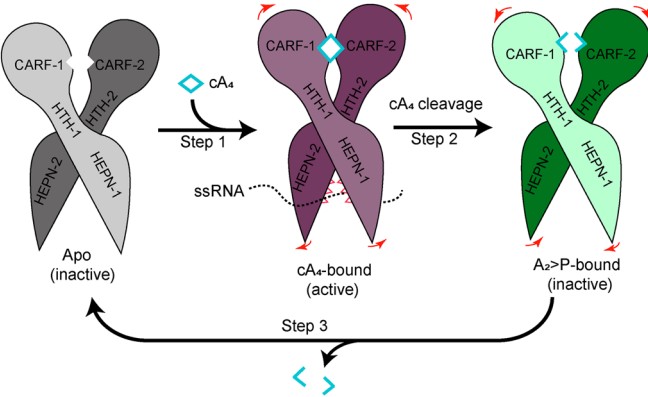

**Figure 6. Schematic representation of the mechanism underlying cA4-mediated allosteric activation of Csm6 ribonuclease.**

Binding of cA4 (depicted as a quadrangular shape) to the CARF domain triggers a transition from an open to a closed conformation (Step 1). The conformational change is transmitted from the CARF domain to the HTH domain, and further extends to the HEPN domain. This leads to the alignment of the R-X$_{4-6}$-H motif (depicted as a jagged shape), resulting in the degradation of ssRNA. Following cA4 cleavage by the ring nuclease activity of the CARF domain, Csm6 adopts a relaxed conformation similar to its apo form (Step 2), rendering the R-X$_{4-6}$-H motif misaligned and catalytically incompetent. Furthermore, subsequent release of the cleaved product (A$_2$ > P) restores Csm6 to its apo state (Step 3), preparing it to bind a new cA4 molecule and initiate the next catalytic cycle.

the Csm6 CARF domain also catalyzes the conversion of cA4 into two A$_2$ > P molecules. This turns Csm6 protein into a relaxed conformation, similar to its apo state (Fig. 5D, step 2; Movie EV2). Specifically, the cleavage of cA4 triggers the opening of CARF domain, promotes the disassociation of the two HTH domain, and subsequently drives the movement of HEPN domain in the opposite direction compared to the previous step. This results in the conversion of R-X$_{4-6}$-H motif into its inactive state (Fig. 5D; Movie EV2). Furthermore, following the release of cleaved product (A$_2$ > P), the CARF domain of Csm6 returns to its apo state and readily accept a new cA4 molecule to start the next catalytic cycle (Fig. 6, step 3). Considering the structural similarity with Crn1 standalone nucleases such as Sso2081, TtCsm6 may employ a stepwise mechanism for cA4 cleavage as observed for Sso2081 (Du et al, 2023). Interestingly, a previous study has demonstrated that not only cA4 but also its linear cleavage intermediate, A$_4$ > P, can activate the ribonuclease activity of TtCsm6 (Niewoehner et al, 2017). Although we did not directly observe the binding of A$_4$ > P in our study, it is likely that A$_4$ > P binding might induce an active conformation in the HEPN domain, similar to the effect of cA4 binding.

It is important to note that all HEPN nucleases studied so far require dimerization to be active (Pillon et al, 2021). Due to the challenge in capturing the structures of TtCsm6 with its HEPN domain bound to RNA substrates, it still remains unclear how the two copies of HEPN motifs cooperate to cleave RNA. It has been suggested that one R-X$_{4-6}$-H copy might be crucial for RNA cleavage, while the other could play a role in positioning the RNA (Pillon et al, 2020). Another possibility is that the HEPN active site could be a composite of the two motifs, where, for example, R$_1$ from one motif and H$_6$ from the other copy contributes to the overall function (Pillon et al, 2020).

While our manuscript was under revision, McQuarrie et al reported a novel activation mechanism of *Streptococcus thermophilus* (St) Csm6 by cA6 (McQuarrie et al, 2023). Their results showed that cA6 binding triggers a 60° rotation between the CARF and HEPN domains, resulting in the opening of the HEPN domain and a repositioning of active site residues. In contrast, structural studies on ToCsm6 have shown that cA4 binding only induces a slight local conformational change in the HEPN domain (Jia et al, 2019; Molina et al, 2019). These findings thus suggest that Csm6 proteins from different species can have distinct activation mechanisms. Indeed, there are obvious differences in both structure and function among these proteins. For instance, ToCsm6 lacks a HTH domain, and its two monomers do not intertwine but instead dimerize along the axis spanning from the CARF to the HEPN domains (Appendix Fig. S7B) (Jia et al, 2019). Moreover, the CARF domain of SisCsx1 does not possess a ring nuclease activity, and a homo-hexamer is essential for its ribonuclease activity in the HEPN domain (Appendix Fig. S7C) (Molina et al, 2019). On the contrary, EiCsm6 exhibits a significant structural similarity to TtCsm6 (Appendix Fig. S7D) (Garcia-Doval et al, 2020). While the structure of apo-EiCsm6 has yet to be determined, it is conceivable that the allosteric activation mechanism observed in TtCsm6 may be applicable to EiCsm6.

Recently, the Csm6 proteins have been applied for highly sensitive viral RNA detection by combining with Cas13 (SHERLOCKv2 and FIND-IT) (Gootenberg et al, 2018; Liu et al, 2021), and for metal ion detection using the DNAzyme-Csm6 tandem assays (Yang et al, 2022), by taking advantage of the cOA-Csm6 signal amplification. However, their detection sensitivities are limited due to the rapid depletion of the linear activator A$_4$ > P, catalyzed by the CARF domain ring nuclease activity of Csm6. Therefore, the hyperactive TtCsm6 variant Y161A identified herein, with its sustained cA4-activated ribonuclease activity, demonstrates great promise in enhancing the sensitivity and specificity of these technologies.

In summary, our work provides a novel allosteric activation mechanism of Csm6 ribonuclease by a cOA second messenger. It enhances our understanding of the cOA signaling pathway of the type III CRISPR-Cas systems in bacteria and archaea immunity. Moreover, the identification of the hyperactive TtCsm6 variant in this study holds great promise for expanding the applications of CRISPR-Cas proteins in genome editing and diagnostics. Future studies will be needed to further elucidate the mechanisms by which Csm6 proteins recognize and cleave RNA substrates in its HEPN domain.

## Methods

### Protein expression and purification

The cDNA of TtCsm6 (GenBank ID: 81626039) was synthesized by GenScript Corporation (Nanjing, China). It was then subcloned into a modified pET bacterial expression vector with an N-terminal 6 × His tag. The variants were generated using the QuickChange-Site-Directed mutagenesis kit, following the manufacturer's instructions (Agilent Technologies).

Protein expression was carried out in *E. coli* Rosetta (DE3) cells, which were induced with 0.5 mM isopropyl β-D-1-thiogalactopyranoside

(IPTG) overnight at 18 °C. Subsequently, the cells were disrupted using French Pressure (Union Biotech, China) in lysis buffer consisting of 20 mM Tris-HCl pH 8.5, 200 mM NaCl, 5% glycerol, and 0.1% Tween-20, supplemented with 10 mM imidazole. The $His_6$-tagged TtCsm6 protein was collected using Ni-NTA resin (Union Biotech, China). After thorough washing, the protein was eluted with lysis buffer supplemented with 200 mM imidazole. The 6×His tag was then removed using homemade preScission protease, and the tag-free protein was further purified using 15Q and Superdex 200 10/300 GL columns (GE Healthcare Life Sciences). The purified protein was concentrated in 20 mM Tris-HCl pH 8.5 and 150 mM NaCl, and stored at −80 °C. All TtCsm6 variants were expressed and purified in a similar manner as described above.

## Crystallization, data collection, and structure determination

Crystallizations were performed at 25 °C using the method of hanging-drop vapor diffusion by mixing reservoir solutions with equal volume of TtCsm6 protein alone or in complex $cA_4$. All the crystals were cryo-protected by the reservoir solutions supplemented with 10-25% glycerol prior to data collection.

For the crystallization of TtCsm6-$A_2 > P$ complex, the TtCsm6 proteins (12 mg/mL) were pre-incubated with $cA_4$ at a molar ratio of 1:1.2 for 3 h at room temperature, and the crystals were grown in a reservoir solution containing 0.2 M Magnesium chloride hexahydrate, 0.1 M Tris pH 8.5 and 25% w/v polyethylene glycol 3350. In the crystallization of TtCsm6 in complex with $cA_4$, the TtCsm6$^{Y161A}$ mutant protein (10 mg/ml) was pre-incubated with $cA_4$ at 4 °C for 30 min, using a molar ratio of 1:1.2, and the crystals were obtained in the reservoir solution containing 0.2 M lithium sulfate, 0.1 M phosphate-citrate pH 4.2 and 20% w/v polyethylene glycol 1000. The apo-TtCsm6 crystals were obtained by mixing the TtCsm6$^{K137A}$ protein (13 mg/mL) with a reservoir solution containing 0.1 M Magnesium chloride hexahydrate, 0.1 M HEPES sodium pH 7.0 and 15% w/v Polyethylene glycol 3350.

X-ray diffraction data of TtCsm6$^{K137A}$ and TtCsm6/$A_2 > P$ complex were collected at BL10U2 with a wavelength of 0.978 Å, while the data collection for TtCsm6$^{Y161A}$/ $cA_4$ complex was conducted at BL18U1 with a wavelength of 0.979 Å, at National Facility for Protein Science in Shanghai (NFPS), at Shanghai Synchrotron Radiation Facility (SSRF). XDS and HKL3000 software were used to process the diffraction data.

Structure determination was performed by molecular replacement with the Phaser-MR program in the PHENIX package (Liebschner et al, 2019), using the structure of apo-TtCsm6 (PDB: 5FSH) as the searching model. Iterative manual model building and structural refinements were conducted using Phenix (Liebschner et al, 2019) and WinCoot (Emsley et al, 2010) software. The final model was validated by MolProbity (Chen et al, 2010). Data collection and structure refinement statistics are summarized in Table 1. All structural figures in this study are generated with the program PyMOL (http://www.pymol.org/).

## cA₄ cleavage assay

A $cA_4$ cleavage assay was conducted to determine the ring nuclease activity of TtCsm6 CARF domain as previously described (Du et al, 2023). Briefly, in a reaction of 50 μL, 2 μM or indicated

concentrations of TtCsm6 were incubated with 40 μM $cA_4$ (Biolog Life Science Institute, Germany) in the buffer containing 20 mM Tris-HCl, pH 8.5, and 50 mM NaCl at 37 °C for 30 min or indicated time. The reaction was quenched by the addition of 50 μL chloroform-isoamylol (24:1), and the nucleotides were extracted for further LC-MS analyses.

## LC-MS analyses

LC-MS analyses were performed as previously described (Du et al, 2023). Briefly, the extracted $cA_4$ and its cleavage products were separated using an HPLC system (LC-20A, Shimadzu) equipped with an RX-C18 column (2.1 × 100 mm, 5 μm) (Zhongpu Science). After injecting 20 μl of the sample, the column was eluted with a linear gradient of buffer-B (acetonitrile with 0.01% TFA) against buffer-A (water with 0.01% TFA) at a flow rate of 0.35 ml/min. The column temperature was set at 40 °C, and ultraviolet (UV) data were collected at a wavelength of 259 nm. The MS data were acquired in negative-ion mode with a scan range of m/z 150–1500 using an Agilent 6520 ACURATE-Mass Q-TOF mass spectrometer. The instrument was operated in both full scan and multiple reaction monitoring (MRM) modes. The temperature was kept at 350 °C, and the capillary voltage was set to 3.5 kV. The nebulizer pressure was adjusted to 40 psi, and the drying gas flow rate was maintained at 10 l/min. Nitrogen was used as both the nebulizer and auxiliary gas.

## Microscale thermophoresis (MST) binding assay

MST experiments were carried out to determine the binding affinities of TtCsm6 and its variants with $cA_4$, following a previously established protocol (Du et al, 2023). All the purified recombinant proteins were dialyzed into MST buffer containing 20 mM HEPES pH 7.4 and 100 mM NaCl prior to the experiment. Then, 90 μl of each protein at a concentration of ~10 μM was mixed and incubated with 300 μM fluorescent dye RED-NHS (MO-L011, NanoTemper Technologies) at room temperature for 30 min in darkness. Following incubation, the labeling reaction was passed through column-B to facilitate the exchange into phosphate-buffered saline at pH 7.4, supplemented with 0.05% Tween-20 (PBST). The labeled proteins (50 nM) were then subjected to incubation with 16 different serial concentrations of $cA_4$ at room temperature for 15 min in PBST. Subsequently, the samples were loaded into capillaries (MO-K022, NanoTemper Technologies). MST was performed using a Monolith NT.115 instrument, employing 60–80% light-emitting diode power and medium MST power settings at 25 °C. All experiments were performed for at least three replicates, and data were analyzed using MO.Affinity Analysis v.2.3 software (NanoTemper Technologies, München, Germany). All figures were generated by using GraphPad Prism 6.0.

## Ribonuclease activity assays

The ribonuclease activities of TtCsm6 and its variants were assessed using both gel- and FRET-based assays. In the gel-based assay, 500 nM of TtCsm6 or its variants were incubated with 250 nM of FAM-labeled RNA, ACUGCAACGCAAUAUACCAUAGCU (Niewoehner and Jinek, 2016), in the presence or absence of 2.5 μM $cA_4$

in the nuclease buffer (25 mM Tris pH 8.5, 50 mM NaCl, 25 mM EDTA) at 37 °C for 60 min. The resulting reaction mixtures were then separated on 10% polyacrylamide-urea denaturing gel, which were imaged and analyzed using the ChemDoc Touch imaging system (Bio-Rad).

In the FRET-based assay, 100 nM of TtCsm6 or its mutants were incubated with 500 nM of a synthetic RNA reporter, 5′-Cy3-CCCCC-BHQ-3′ (BioSune Biotechnology, Fuzhou, China), with or without 100 nM of $cA_4$ in the buffer containing 25 mM HEPES pH 6.8, 50 mM NaCl, 10 mM EDTA. The fluorescence intensities were recorded on a microplate reader (BioTek), with excitation and emission wavelengths set as 540 nm and 570 nm, respectively. The reactions were carried out at 37 °C for 120 min, and fluorescent kinetics were measured every 5 min.

## Data availability

Atomic coordinates and structure factors of TtCsm6[K137A], TtCsm6/$A_2 > P$ complex, and TtCsm6/$cA_4$ complex have been deposited with the Protein Data bank (https://www.rcsb.org) under accession codes 8JBC, 8JBB and 8JH1, respectively.

## Peer review information

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

## Acknowledgements

We thank the staffs from BL10U2 and BL18U1 beamlines of National Facility for Protein Science in Shanghai (NFPS) at Shanghai Synchrotron Radiation Facility (SSRF), Shanghai, People's Republic of China, for assistance with X-ray data collection. This work is supported by the National Natural Science Foundation of China 31971222.

## Author contributions

**Liyang Du**: Resources; Data curation; Software; Formal analysis; Validation; Investigation; Visualization; Methodology. **Qinwei Zhu**: Validation; Visualization; Methodology. **Zhonghui Lin**: Conceptualization; Resources; Data curation; Supervision; Funding acquisition; Validation; Writing—original draft; Project administration; Writing—review and editing.

## Disclosure and competing interests statement

The authors declare no competing interests.

