## [Peer Review File · The EMBO Journal]

Molecular mechanism of allosteric activation of the CRISPR ribonuclease Csm6 by cyclic tetra-adenylate

Liyang Du, Qinwei Zhu, and zhonghui lin
DOI: 10.15252/emboj.2023115215

Corresponding author(s): zhonghui lin (zhonghui.lin@fzu.edu.cn)

Review Timeline:

Submission Date:	3rd Aug 23
Editorial Decision:	29th Sep 23
Revision Received:	18th Oct 23
Editorial Decision:	15th Nov 23
Revision Received:	16th Nov 23
Accepted:	27th Nov 23

Editor: Cornelius Schneider

Transaction Report:

Dear Prof. Lin,

Thank you for submitting your manuscript for consideration by the EMBO Journal. It has now been seen by three referees whose comments are shown below.

Given the referees' positive recommendations, I would like to invite you to submit a revised version of the manuscript, addressing the comments of all three reviewers. I should add that it is EMBO Journal policy to allow only a single round of revision, and acceptance of your manuscript will therefore depend on the completeness of your responses in this revised version.

Thank you for the opportunity to consider your work for publication. I look forward to your revision.

Yours sincerely,

Cornelius Schneider, PhD
Editor
The EMBO Journal
c.schneider@embojournal.org

We realize that it is difficult to revise to a specific deadline. In the interest of protecting the conceptual advance provided by the work, we recommend a revision within 3 months (28th Dec 2023). Please discuss the revision progress ahead of this time with the editor if you require more time to complete the revisions. Use the link below to submit your revision:

Referee #1:

Du et al. report a structural study into the cA4 activated Csm6 ancillary ribonuclease from *Thermus thermophilus* (Tth). Crystal structures are presented in the apo-, cA4-activated and post cA4 cleavage states. The latter structure is very similar to the previously published apo-structure, with differences primarily in the loops of the CARF domains that enclose the ligand. Ring nuclease activity is observed and probed by sdm - giving results that are largely confirmatory to the literature. Some mutations in the cA4 binding site led to prolonged activation of the nuclease and allowed crystallisation in the presence of cA4, yielding an "active" structure for the first time for this enzyme, albeit at medium resolution. This allows a detailed exploration of the structural changes that accompany cA4 binding, leading to activation of the HEPN domain by relatively subtle structural reorganisation.

The work is carefully executed and interpreted. The data support the conclusions and the figures are clear and well presented. Overall, this study adds to our understanding of the structural basis for activation and deactivation of Csm6 family enzymes - although considerable diversity exists in this family. The identification of a hyperactive mutant enzyme may, as the authors suggest, be useful for diagnostic assays that use Csm6 to amplify signal.

Overall, this manuscript reports new findings of some novelty, albeit the major conclusions are not particularly surprising given previous work on Csm6 family proteins.

Malcolm White

Specific points:

1. Figure 3. Where errors are in the range 30-50% it is not appropriate to report KD values to three significant figures (eg 80.0 {plus minus} 46.4). Please round off.

Referee #2:

The study involves the investigation of CRISPR ribonuclease Csm6 from *Thermus thermophilus* and its allosteric activation by cyclic tetra adenylate (cA4). The authors illustrated the crystal structures of TtCsm6 with cA4 bound in multiple catalytic states. Residues contributing to cA4 recognition/cleavage were identified. TtCsm6Y161A, which exhibited a strong binding affinity to cA4 but lacks cleavage ability, was employed to obtain structures representing the pre-cleavage state of cA4 bound structure. When comparing the apo-, pre-cleavage and post cleavage states, the authors unveiled an allosteric activation of Csm6 by cA4, including ligand binding induced HTH domain dimerization and significant conformational change of HEPN domain, particularly in the R-X4-6-H motif. Additionally, a mobile loop (165-179), switching between its open (apo- or cA4 post-cleavage) and close (cA4 pre-cleavage) states, was shown to mediate the allosteric regulation of HEPN. The research described in the manuscript is interesting, however there are following significant concerns that need to be resolved before it can be published.

Major concerns:

1. Evidence might be needed to confirm that the majority of cA4 in the MST binding assays remained uncleaved after pre-incubation with Csm6 for 15 minutes at room temperature. This should be relevant even though concentration of Csm6 used in the MST assays was ~50nM (Fig 3) as compared to 2uM used in the kinetic plot (Supplementary Fig. 1D), where 40% of cA4 (40 μ M) were cleaved after being incubated for 15min at 37 {degree sign}C. Also, details of the MST measurements are not provided in supplementary data; apparently low MST response amplitudes are concerning and should be discussed if true.
2. It is questionable that T81A showed 100-fold weaker cA4 binding affinity than WT, but surprisingly performed "more significant and prolonged RNA degradation". This contrasts with other mutants such as T59A/S60A which show less drastic drop in binding affinity but exhibit no RNA hydrolysis. A thorough discussion is needed for this unusual result.
3. Given the importance ascribed by authors to the mobile loop, and the fact that their single point mutations did not significantly affect cA4 hydrolysis, why weren't multiple point mutations tried?
4. It is not discussed why K137 (and not N416 and H422, or even WT for instance) was hypothesized to overcome the potential

side-effects of Ni binding in the HEPN domain of the previous TtCsm6 structure. Is there an alternate explanation/discussion or follow up from the lack of effect of Ni binding in the K137A structure.

5. As cA4 binding and cleavage are two distinct mechanisms, roles of residues need to be summarized. For instance, given the data from cA4 cleavage assay and MST, T59/S60, T81 and K137 are likely responsible for cA4 binding, while Y161 and N164 mostly contribute to cA4 cleavage.

6. In the absence of an RNA bound Csm6 (/HEPN) structure, could authors dock/model the RNA to further clarify the mechanism of how the conformational change described in this manuscript contributes to the activation of HEPN?

7. Given that many CARFs, such as ToCsm6, perform a stepwise cleavage of cA4, featuring an intermediate ApApAp>p, were any similar products observed in this study? If no, a discussion of either a lack of information on these states and/or of how authors view/predict their post cleavage state to be affected (or not) by ApApAp>p binding could help provide full picture.

Minor concerns:

1. Details are required for the LC-MS methods, such as buffer conditions and column dimensions. The methods of site-directed mutagenesis are missing.
2. Csm6 is a non-specific RNase; is there a reason to choose a particular sequence in ribonuclease activity assay? An explanation or discussion would help.
3. Error bars for the kinetic plot in Supplementary Figure 1D were not shown. At least the reason should be provided in the text.

Referee #3:

Zhonghui Lin and coworkers report on the structure of *Thermus thermophilus* dimeric Csm6 bound to ApA>P and cA4 and compare it with the published structure in the apo state. They conclude that cA4 is converted to ApA>p in the CARF pocket over time, thereby shutting off the nuclease activity of the HEPN domain, a feature reported previously in the literature. The novel feature of the paper is the use of a hyperactive mutant of TtCsm6, namely Y165A, to trap uncleaved cA4 within the CARF pocket. This has allowed them to compare and contrast the structures of Csm6 in the apo, ApA>p and cA4 bound states, thereby identifying the allosteric conformational changes in TtCsm6, particularly in the HEPN domains, on cA4 complex formation. The new insights reported in this paper merits its publication.

I would request one clarification.

There remains a challenge related to solving the structure of the cA4-dimeric TtCsm6 complex with bound RNA within the HEPN pocket. The authors note that the pair of HEPN domains of TtCsm6 undergo a conformational change to form a composite binding pocket in the cA4 bound state. In their view, do separate ssRNAs bind to each HEPN domain in the TtCsm6 dimer, or alternately a single ssRNA is positioned in the composite pocket to mediate cleavage.

A point-by-point response to the referees' comments

Referee #1:

*Du et al. report a structural study into the cA₄ activated Csm6 ancillary ribonuclease from *Thermus thermophilus* (Tth). Crystal structures are presented in the apo-, cA₄-activated and post cA₄ cleavage states. The latter structure is very similar to the previously published apo-structure, with differences primarily in the loops of the CARF domains that enclose the ligand. Ring nuclease activity is observed and probed by sdm - giving results that are largely confirmatory to the literature. Some mutations in the cA₄ binding site led to prolonged activation of the nuclease and allowed crystallisation in the presence of cA₄, yielding an "active" structure for the first time for this enzyme, albeit at medium resolution. This allows a detailed exploration of the structural changes that accompany cA₄ binding, leading to activation of the HEPN domain by relatively subtle structural reorganisation.*

The work is carefully executed and interpreted. The data support the conclusions and the figures are clear and well presented. Overall, this study adds to our understanding of the structural basis for activation and deactivation of Csm6 family enzymes - although considerable diversity exists in this family. The identification of a hyperactive mutant enzyme may, as the authors suggest, be useful for diagnostic assays that use Csm6 to amplify signal.

Overall, this manuscript reports new findings of some novelty, albeit the major conclusions are not particularly surprising given previous work on Csm6 family proteins.

Malcolm White

Response: We appreciate the reviewer for the positive comments on our manuscript.

Specific points:

1. Figure 3. Where errors are in the range 30-50% it is not appropriate to report K_D values to three significant figures (eg 80.0 {plus minus} 46.4). Please round off.

Response: We thank the reviewer for raising this point and have corrected the K_D values accordingly.

Referee #2:

*The study involves the investigation of CRISPR ribonuclease Csm6 from *Thermus thermophilus* and its allosteric activation by cyclic tetra adenylate (cA₄). The authors illustrated the crystal structures of TtCsm6 with cA₄ bound in multiple catalytic states.*

Residues contributing to cA₄ recognition/cleavage were identified. TtCsm6Y161A, which exhibited a strong binding affinity to cA₄ but lacks cleavage ability, was employed to obtain structures representing the pre-cleavage state of cA₄ bound structure. When comparing the apo-, pre-cleavage and post cleavage states, the authors unveiled an allosteric activation of Csm6 by cA₄, including ligand binding induced HTH domain dimerization and significant conformational change of HEPN domain, particularly in the R-X4-6-H motif. Additionally, a mobile loop (165-179), switching between its open (apo- or cA₄ post-cleavage) and close (cA₄ pre-cleavage) states, was shown to mediate the allosteric regulation of HEPN. The research described in the manuscript is interesting, however there are following significant concerns that need to be resolved before it can be published.

Response: We thank the reviewer for the overall positive comments. We have performed additional experiments and revised part of the statements to address the reviewer's comments. As a result, the paper has been greatly improved.

Major concerns:

- 1. Evidence might be needed to confirm that the majority of cA₄ in the MST binding assays remained uncleaved after pre-incubation with Csm6 for 15 minutes at room temperature. This should be relevant even though concentration of Csm6 used in the MST assays was ~50nM (Fig 3) as compared to 2uM used in the kinetic plot (Supplementary Fig. 1D), where 40% of cA₄ (40 μM) were cleaved after being incubated for 15min at 37 {degree sign}C. Also, details of the MST measurements are not provided in supplementary data; apparently low MST response amplitudes are concerning and should be discussed if true.*

Response: We appreciate the reviewer for these insightful comments.

In response to the comment regarding the cleavage status of cA₄ in the MST binding assays, we conducted additional experiments to confirm the cleavage activity of cA₄ under MST conditions. Specifically, we pre-incubated 50 nM TtCsm6 protein with 4 μM cA₄, the maximal concentration used in the MST experiment, at room temperature for 15 minutes in a phosphate buffered saline solution at pH 7.4, supplemented with 0.05% tween-20. Subsequently, we assessed the efficiency of cA₄ cleavage for both the wild-type and mutant TtCsm6 proteins using HPLC. The results, as shown in Supplementary Fig. 4 of the revised manuscript, unequivocally demonstrate that cA₄ remains uncleaved after pre-incubation with both wild-type and mutant TtCsm6 for 15 minutes at room temperature in the MST binding conditions (P7 line 20-22).

Regarding the MST response amplitudes, it is generally recognized that a reliable MST signal should exhibit a response amplitude of more than 5 units, which signifies a substantial difference between the bound and unbound states. We have shown that all the response amplitudes obtained in our experiments exceeded this threshold, affirming the high quality of our binding data. We have provided a more detailed description of the MST measurements in the revised manuscript (P21 line 21-28, P22

line 1-10).

2. *It is questionable that T81A showed 100-fold weaker cA₄ binding affinity than WT, but surprisingly performed "more significant and prolonged RNA degradation". This contrasts with other mutants such as T59A/S60A which show less drastic drop in binding affinity but exhibit no RNA hydrolysis. A thorough discussion is needed for this unusual result.*

Response: We appreciate the reviewer's insightful comments regarding the T81A mutation. It is indeed intriguing that the T59A/S60A mutant exhibits only about a 5-fold weaker binding affinity than the wild-type (WT) yet is completely impaired in cA₄ cleavage and RNA degradation. In contrast, the T81A mutant displays approximately a 100-fold weaker cA₄ binding affinity than WT but surprisingly demonstrates more pronounced and sustained RNA degradation.

We propose a possible explanation for this seemingly paradoxical result. T59 and S60 are known to play a crucial role in positioning the 5'-phosphate and 2'-OH nucleophile. Therefore, the T59A/S60A mutation likely disrupts the proper positioning of cA₄ within the CARF domain, potentially interfering with the allosteric activation of the HEPN RNase. This disruption, in turn, explains the lack of cA₄ cleavage and RNA degradation observed in this mutant.

Conversely, the T81A mutation, although it leads to a substantial reduction in cA₄ binding affinity, may not significantly alter the binding mode of cA₄ within the CARF domain. As a result, while this mutation initially causes a decrease in RNA degradation within the first 30 minutes of the reaction compared to the WT, over an extended timeframe, the T81A mutant demonstrates more significant and prolonged RNA degradation. The persistence of cA₄ binding within the CARF domain, albeit with reduced affinity, may account for this unexpected observation.

We have included this discussion in the revised manuscript, and hope that would be helpful for better understanding of these intriguing results (P8 line 14-28, P9 line 1-2).

3. *Given the importance ascribed by authors to the mobile loop, and the fact that their single point mutations did not significantly affect cA₄ hydrolysis, why weren't multiple point mutations tried?*

Response: This is a great suggestion. To address this comment, we have carried out additional mutations, including both double and triple mutations. As shown in the revised Fig. 2d, R172A/R173A double mutation caused approximately 50% reduction of cA₄ cleavage, while the Y167A/R172A/R173A triple mutation completely abolished cA₄ cleavage. These results thus further underscore the importance of the mobile loop. We have incorporated these results into the revised version of the manuscript (P7 line 6-10).

4. *It is not discussed why K137 (and not N416 and H422, or even WT for instance) was hypothesized to overcome the potential side-effects of Ni binding in the HEPN domain of the previous TtCsm6 structure. Is there an alternate explanation/discussion or follow up from the lack of effect of Ni binding in the K137A structure.*

Response: We thank the reviewer for raising this important point. The Ni-free structure can be obtained by excluding metal ions from the crystallization solution, and the structure of TtCsm6^{K137A} was initially determined to investigate the impact of this mutation on the integrity of TtCsm6 homodimer. We apologize for any confusion that has caused, and have clarified this point in the revised version of this manuscript (P11 line 11-15).

5. *As cA₄ binding and cleavage are two distinct mechanisms, roles of residues need to be summarized. For instance, given the data from cA₄ cleavage assay and MST, T59/S60, T81 and K137 are likely responsible for cA₄ binding, while Y161 and N164 mostly contribute to cA₄ cleavage.*

Response: We appreciate the reviewer for this constructive suggestion. Based on the results from cA₄ binding and cleavage assays, we propose that residues T81, and K137 are primarily responsible for cA₄ binding, while residues Y161 and N164 play a major role in cA₄ cleavage. Additionally, residues T59 and S60, which coordinate both the scissile phosphate and the 2'-OH nucleophile, may be required for both cA₄ binding and cleavage. We have incorporated these results into the revised version of the manuscript (P8 line 1-5).

6. *In the absence of an RNA bound Csm6 (/HEPN) structure, could authors dock/model the RNA to further clarify the mechanism of how the conformational change described in this manuscript contributes to the activation of HEPN?*

Response: We appreciate the reviewer's valuable input. Investigating the precise mechanism of HEPN-directed RNA cleavage needs high-resolution structures of HEPN domains in complex with RNA substrates. However, capturing these structures is challenging due to the transient nature of such interactions. While a successful example exists with *Thermococcus onnurineus* Csm6 (ToCsm6) complexed with cA₄ (bound in both CARF and HEPN domains), which shed light on a potential adenosine-specific endoribonuclease mechanism, this may not apply to TtCsm6. Unlike ToCsm6, TtCsm6 lacks cA₄ cleavage activity in its HEPN domain. In addition, the two proteins are distinct in the topology of HEPN domain. On the other hand, RNA binding may also induce a conformational change in the R-X₄-H motif of HEPN domain. As such, constructing an RNA-bound TtCsm6 model remains challenging based on our current understanding.

Despite these challenges, we have proposed a potential HEPN activation mechanism based on our crystal structures of TtCsm6 in various catalytic states. In the absence of cA₄, H422 forms a hydrogen bond with N416, and R415 is buried

inside the domain, where it is stabilized by E332. Upon cA₄ binding in the CARF domain, significant conformational changes occur in the R-X₄-H motif, for instance, the hydrogen bonds between N416 and H422, as well as between R415 and E332 were disrupted. This allows R415 and H422 to participate in 5'-phosphate stabilization and 2'-OH nucleophile activation. Our mutagenesis results that E332A and N416A mutation completely abolished HEPN RNase activity support this hypothesis.

We acknowledge the complexity of constructing an RNA-bound model, and hope further structural studies on protein-DNA complexes will enhance our understanding of substrate recognition and cleavage at the molecular level. We have incorporated this discussion into the revised version of the manuscript (P12 line 20-28, P13 line 1-4 and P13 line 21-P14 line 2).

7. *Given that many CARFs, such as TtCsm6, perform a stepwise cleavage of cA₄, featuring an intermediate ApApAp>p, were any similar products observed in this study? If no, a discussion of either a lack of information on these states and/or of how authors view/predict their post cleavage state to be affected (or not) by ApApAp>p binding could help provide full picture.*

Response: We thank the reviewer for this valuable input. Sequence alignments have revealed a high similarity between the CARF domain of TtCsm6 and Crn1 standalone ring nucleases such as Sso2081, particularly in the signature motif-I (GTS) and motif-II (TxGxK). This observation suggests that TtCsm6 may indeed employ a stepwise mechanism for cA₄ cleavage as observed for Sso2081. Interestingly, previous study has demonstrated that not only cA₄ but also the linear cA₄ cleavage intermediate, ApApAp>p (A₄>P), can activate the ribonuclease activity of TtCsm6 (Niewoehner *et al. Nature*, 2017). Although we did not directly observe the binding of A₄>P in our study, it is likely that A₄>P binding might induce an active conformation in the HEPN domain, similar to the effect of cA₄ binding. We have included this discussion in the revised version of the manuscript to offer a more comprehensive view of our findings (P13 line 14-20).

Minor concerns:

1. *Details are required for the LC-MS methods, such as buffer conditions and column dimensions. The methods of site-directed mutagenesis are missing.*

Response: We thank the reviewer's suggestion, and have included more detailed descriptions of the methods for LC-MS and site-directed mutagenesis in the revised manuscript (P19 line 5-7, P21 line 6-19).

2. *Csm6 is a non-specific RNase; is there a reason to choose a particular sequence in ribonuclease activity assay? An explanation or discussion would help.*

Response: We thank the reviewer for pointing out this important question. The sequence specificity of TtCsm6 has been carefully analyzed by Niewoehner *et al.*

Their investigation clearly demonstrated that TtCsm6 does not exhibit a preference for a specific substrate (Niewoehner *et al. RNA*, 2016). The RNA substrate used in this study was randomly generated as previously described (Niewoehner *et al. RNA*, 2016).

3. Error bars for the kinetic plot in Supplementary Figure 1D were not shown. At least the reason should be provided in the text.

Response: The data in Supplementary Fig. 1d was obtained from a series of time points, and each time point was tested once. We thank the reviewer for raising this point and have clarified this point in the corresponding figure legend.

Referee #3:

Zhonghui Lin and coworkers report on the structure of Thermus thermophilus dimeric Csm6 bound to ApA>P and cA₄ and compare it with the published structure in the apo state. They conclude that cA₄ is converted to ApA>p in the CARF pocket over time, thereby shutting off the nuclease activity of the HEPN domain, a feature reported previously in the literature. The novel feature of the paper is the use of a hyperactive mutant of TtCsm6, namely Y165A, to trap uncleaved cA₄ within the CARF pocket. This has allowed them to compare and contrast the structures of Csm6 in the apo, ApA>p and cA₄ bound states, thereby identifying the allosteric conformational changes in TtCsm6, particularly in the HEPN domains, on cA₄ complex formation. The new insights reported in this paper merits its publication.

Response: We thank the reviewer for the positive comments.

I would request one clarification. There remains a challenge related to solving the structure of the cA₄-dimeric TtCsm6 complex with bound RNA within the HEPN pocket. The authors note that the pair of HEPN domains of TtCsm6 undergo a conformational change to form a composite binding pocket in the cA₄ bound state. In their view, do separate ssRNAs bind to each HEPN domain in the TtCsm6 dimer, or alternately a single ssRNA is positioned in the composite pocket to mediate cleavage.

Response: We appreciate the reviewer's insightful comment. Based on the structural and biochemical data available for HEPN family members, it is suggested that the dimeric HEPN domain accommodates one ssRNA in a catalytic cycle. This notion is supported by the recent structure of *Thermococcus onnurineus* Csm6 (ToCsm6) complexed with cA₄, where both the CARF and HEPN domains separately bind one cA₄ molecule (Jia *et al. Mol Cell*, 2019), and its HEPN domain accommodates only a single phosphodiester bond from cA₄.

It's important to note that all HEPN nucleases studied so far require dimerization to be active (Pillon *et al. Crit Rev Biochem Mol Biol*, 2021). It has been suggest that one R-X₄-H copy might be crucial for RNA cleavage, while the other could play a

role in positioning the RNA (Pillon *et al. J. Biol. Chem.*, 2020). Another possibility is that the HEPN active site could be a composite of the two motifs, where, for example, R₁ from one motif and H₆ from the other copy contribute to the overall function.

We have incorporated this discussion into the revised version of the manuscript (P13 line 21-P14 line 2).

Dear Dr Lin,

Thank you for submitting a revised version of your manuscript. Your study has now been seen by all original referees, who find that their previous concerns have been addressed and now recommend publication of the manuscript. In addition, there remain a few mainly editorial points that must be addressed before I can extend formal acceptance of the manuscript:

1. Please rename the conflict-of-interest statement to "DISCLOSURE AND COMPETING INTERESTS STATEMENT"
2. CRedit has replaced the traditional author contributions section because it offers a systematic, machine-readable author contributions format that allows for more effective research assessment. Please remove the Authors Contributions from the manuscript and use the free text boxes beneath each contributing author's name in our online submission system to add specific details on the author's contribution. More information is available in our guide to authors.
3. Appendix Figure S7 is not referenced in the manuscript.
4. Please take another look at the source data checklist and complete the missing sections.
5. Please add page numbers to the Appendix Figures file and reference them in the ToC section within the same file.
6. Please make sure that URLs for 8JBC, 8JBB and 8JH1 are provided and available.
7. Please rename the movie files to Movie EV1-EV2 with the corresponding callouts, remove their legends from the ms files, and add them into a zipped file together with each movie file.
8. Please make sure that the order of the sections in the manuscript is as follows: abstract, introduction, results, discussion, materials & methods, data availability section, acknowledgments, disclosure statement and competing interests, references, main figure legends, tables, expanded figure legends.

9. Synopsis:

Papers published in The EMBO Journal are accompanied online by a 'Synopsis' to enhance discoverability of the manuscript. It consists of A) a short (1-2 sentences) summary of the findings and their significance, B) 3-4 bullet points highlighting key results and C) a synopsis image that is 550x300-600 pixels large (width x height, jpeg or png format). You can either show a model or key data in the synopsis image. Please note that the image size is rather small, and that text needs to be readable at the final size. Please send us this information together with the revised manuscript.

Thank you again for giving us the chance to consider your manuscript for The EMBO Journal. I look forward to receiving the final version. Please note that manuscripts accepted and exported after Friday 17th of November will be published online in January 2024.

With best regards,

Cornelius Schneider

Cornelius Schneider, PhD
Editor
The EMBO Journal
c.schneider@embojournal.org

- a point-by-point response to the referees' comments, with a detailed description of the changes made (as a word file).
- a word file of the manuscript text.
- individual production quality figure files (one file per figure)
- a complete author checklist, which you can download from our author guidelines

(<https://www.embopress.org/page/journal/14602075/authorguide>).
- Expanded View files (replacing Supplementary Information)
Please see out instructions to authors
<https://www.embopress.org/page/journal/14602075/authorguide#expandedview>

We realize that it is difficult to revise to a specific deadline. In the interest of protecting the conceptual advance provided by the work, we recommend a revision within 3 months (13th Feb 2024). Please discuss the revision progress ahead of this time with the editor if you require more time to complete the revisions. Use the link below to submit your revision:

Referee #1:

The authors have dealt with all my concerns.

Referee #2:

The authors have demonstrated an exceptional level of diligence and scientific rigor in their responses to the initial review, showcasing a commitment to advancing the clarity and robustness of their study. Their meticulous approach is evident in the additional experiments conducted to validate the cleavage status of cA4 in MST binding assays, a crucial step that enhances the reliability of their findings. The comprehensive discussion surrounding the perplexing result of the T81A mutant not only highlights the authors' keen analytical skills but also underscores their commitment to providing a thorough and convincing interpretation of their data. The authors have further underscored the significance of the mobile loop through additional mutations, including double and triple mutations, showcasing its crucial role in cA4 cleavage. Moreover, the authors' responsiveness is further demonstrated in the detailed descriptions of LC-MS methods and site-directed mutagenesis, addressing potential concerns and ensuring the reproducibility of their methods. The clarification regarding the non-specific nature of TtCsm6 as a ribonuclease and the thoughtful explanation of the absence of error bars in the kinetic plot exemplify the authors' transparency and commitment to scientific integrity. Additionally, the enriched discussion on the roles of specific residues in cA4 binding and cleavage, coupled with the acknowledgment of the challenges in constructing an RNA-bound model, showcases the authors' dedication to advancing the understanding of CRISPR-Cas systems. The proposal of a potential HEPN activation mechanism and the inclusion of a discussion on the stepwise mechanism for cA4 cleavage and intermediate products further elevate the manuscript, offering a more comprehensive and insightful perspective.

In light of these commendable efforts, I wholeheartedly endorse the publication of this manuscript, recognizing its substantial contribution to the current knowledge of CRISPR-Cas systems and the intricate regulation of Csm6 ribonucleases.

Referee #3:

I recommend publication of the revised version.

A point-by-point response to the referees' comments

1. Please rename the conflict-of-interest statement to "DISCLOSURE AND COMPETING INTERESTS STATEMENT"

Response: The conflict-of-interest statement has been renamed to "DISCLOSURE AND COMPETING INTERESTS STATEMENT".

2. CRediT has replaced the traditional author contributions section because it offers a systematic, machine-readable author contributions format that allows for more effective research assessment. Please remove the Authors Contributions from the manuscript and use the free text boxes beneath each contributing author's name in our online submission system to add specific details on the author's contribution. More information is available in our guide to authors.

Response: The "Authors Contributions" section was removed from the manuscript, was instead described in the online submission system.

3. Appendix Figure S7 is not referenced in the manuscript.

Response: We apologize for this oversight. We have added a brief description (highlighted in blue color) and referenced Appendix Figure S7 in the DISCUSSION section of the revised manuscript.

4. Please take another look at the source data checklist and complete the missing sections.

Response: We have checked the source data checklist, and confirmed that all data have been provided.

5. Please add page numbers to the Appendix Figures file and reference them in the ToC section within the same file.

Response: Page numbers have been added to the Appendix Figures file, and referenced in the ToC section within the same file.

6. Please make sure that URLs for 8JBC, 8JBB and 8JH1 are provided and available.

Response: URLs for 8JBC, 8JBB and 8JH1 have been provided in the DATA AVAILABILITY section.

7. Please rename the movie files to Movie EV1-EV2 with the corresponding callouts, remove their legends from the ms files, and add them into a zipped file together with each movie file.

Response: The movie files have been renamed to Movie EV1-EV2 with the corresponding callouts, and their legends have been moved from the ms files into a zipped file together with each movie file.

8. Please make sure that the order of the sections in the manuscript is as follows: abstract, introduction, results, discussion, materials & methods, data availability

section, acknowledgments, disclosure statement and competing interests, references, main figure legends, tables, expanded figure legends.

Response: The sections have been reordered accordingly in the revised manuscript.

9. Synopsis: Papers published in The EMBO Journal are accompanied online by a 'Synopsis' to enhance discoverability of the manuscript. It consists of A) a short (1-2 sentences) summary of the findings and their significance, B) 3-4 bullet points highlighting key results and C) a synopsis image that is 550x300-600 pixels large (width x height, jpeg or png format). You can either show a model or key data in the synopsis image. Please note that the image size is rather small, and that text needs to be readable at the final size. Please send us this information together with the revised manuscript.

Response: We have provided a 'Synopsis' which includes :

- A) a short summary of the findings and significance;
- B) 4 bullet points highlighting key results;
- C) a synopsis image with 550x300-600 pixels large (width x height, jpeg format).

Dear Prof. Lin,

I am pleased to inform you that your manuscript has been accepted for publication in the EMBO Journal.

Yours sincerely,

Cornelius Schneider, PhD
Editor
The EMBO Journal
c.schneider@embojournal.org
